# PROTACs in the Management of Prostate Cancer

**DOI:** 10.3390/molecules28093698

**Published:** 2023-04-25

**Authors:** Poornachandra Yedla, Ahmed O. Babalghith, Vindhya Vasini Andra, Riyaz Syed

**Affiliations:** 1Department of Pharmacogenomics, Institute of Translational Research, Asian Healthcare Foundation, Asian Institute of Gastroenterology Hospitals, Gachibowli, Hyderabad 500082, India; 2Department of Medical Genetics, Faculty of Medicine, Umm Al-Qura University, Makkah 21955, Saudi Arabia; 3Department of Medical Oncology, Omega Hospitals, Gachibowli, Hyderabad 500032, India; 4Department of Chemiinformatics, Centella Scientific, JHUB, Jawaharlal Nehru Technological University, Hyderabad 500085, India

**Keywords:** PROTAC, prostate cancer, androgen receptor, mutation, resistance

## Abstract

Cancer treatments with targeted therapy have gained immense interest due to their low levels of toxicity and high selectivity. Proteolysis-Targeting Chimeras (PROTACs) have drawn special attention in the development of cancer therapeutics owing to their unique mechanism of action, their ability to target undruggable proteins, and their focused target engagement. PROTACs selectively degrade the target protein through the ubiquitin–proteasome system, which describes a different mode of action compared to conventional small-molecule inhibitors or even antibodies. Among different cancer types, prostate cancer (PC) is the most prevalent non-cutaneous cancer in men. Genetic alterations and the overexpression of several genes, such as FOXA1, AR, PTEN, RB1, TP53, etc., suppress the immune response, resulting in drug resistance to conventional drugs in prostate cancer. Since the progression of ARV-110 (PROTAC for PC) into clinical phases, the focus of research has quickly shifted to protein degraders targeting prostate cancer. The present review highlights an overview of PROTACs in prostate cancer and their superiority over conventional inhibitors. We also delve into the underlying pathophysiology of the disease and explain the structural design and linkerology strategies for PROTAC molecules. Additionally, we touch on the various targets for PROTAC in prostate cancer, including the androgen receptor (AR) and other critical oncoproteins, and discuss the future prospects and challenges in this field.

## 1. Introduction

Targeted therapies work to treat cancer by impeding certain proteins that promote tumor growth and metastasis throughout the body. These therapies have made significant strides in the treatment of cancer over the past few decades and are now an effective option for cancer patients [1]. As a result, the development of small-molecule inhibitors (SMIs) enables effective therapeutic approaches to target overexpressed proteins in several cancers. However, due to the poor therapeutic efficacy, genetic alterations/mutations in the target protein, drug resistance, and off-target effects of these SMIs treatments, researchers are currently looking for more effective and precise approaches to target the oncoproteins that are associated with cancer progression [2]. One such approach is Proteolysis-Targeting Chimeras (PROTACs), which overcomes the constraints of conventional SMIs and selectively degrades the target protein. Instead of altering the protein’s function by dwelling in an active site, PROTACs use the cell’s own degradation machinery to degrade the target protein [3]. This innovative approach not only makes it possible to create more effective and targeted therapies for cancer and other intractable diseases, but it also opens up hitherto “undruggable” regions of the proteome [4,5]. PROTACs are now acknowledged as a novel approach within drug development and they have the potential to become the next generation of anticancer therapeutics.

Cancer is one of the most intricate and rapidly propagating diseases of the present century with a high mortality rate. Among several cancers, prostate cancer (PC) is the most prevalent urogenital malignancy in men, characterized by aberrant cell division that causes the prostate gland to develop abnormally [6]. The primary cause of PC mortality is metastatic prostate cancer, where cancer cells have metastasized to other regions of the body such as the bone, pelvis, lumbar vertebra, bladder, rectum, and brain [7]. Approximately half (48%) of all incidents of cancer in males are prostate, lung, bronchus, and colorectal cancers; PC alone accounts for 27% of diagnoses. In 2020, there were 1,414,249 newly diagnosed cases and 375,000 deaths globally because of PC, and in 2022, PC cases were projected to total 268,490 cases (14% of all new cancer cases) and 34,500 deaths alone in the United States [8,9]. Somatic and germline genetic aberrations, increased AR expression, gene amplification, activation of CYP17A1 to increase androgen production, and the emergence of AR splice variants are promising contributing factors for PC progression [10]. The current treatment for early stage prostate cancer is androgen deprivation therapy (ADT), which can be achieved either surgically or medically by castration using anti-androgens or luteinizing hormone-releasing hormone (LHRH) agonists or antagonists. Even though ADT results in remissions that last around two to three years, the condition eventually advances to castration-resistant prostate cancer (CRPC), which has a dismal prognosis and significant treatment complications [11].

With the burgeoning clinical requirement and several limitations of SMIs of PC, there has been considerable development in the discovery of anticancer PROTACs that target the various oncoproteins that are involved in disease progression. Due to its distinctive mode of action, PROTAC technology may compensate for the shortcomings of conventional drug therapy, which accelerates its development. Further, PROTAC is a highly promising technology in terms of its clinical applications, since ARV-110, ARV-766, and AR-LDD, which are some of the AR-targeting PROTACs, are in clinical trials for the treatment of PC [12]. In this review, we introduce the principles and development of prostate cancer-targeted PROTAC technology and summarize the application of PROTACs in targeting the crucial proteins that are involved in tumorigenesis. In addition, we summarized the genetic abnormalities (somatic and germline mutations) of different oncoproteins associated with PC development, which may help in the expansion of precision medicine for PC in the near future. 

## 2. Ubiquitination and Proteolysis-Targeting Mechanism

Protein degradation can take on many different forms and is critical for regulating various cellular and biological processes. The ubiquitin–proteasome system (UPS) is responsible for the degradation of most proteins in eukaryotic cells. This pathway involves the covalent attachment of the highly conserved 76 amino acid protein ubiquitin onto target proteins prior to their recognition and degradation by the 26S proteasome [13,14]. The cellular protein degradation process is necessary for optimal cellular activity, including proliferation, differentiation, and cell death. One such important pathway for post-translational protein regulation is ubiquitin-dependent proteolysis. Ubiquitin-activating enzymes (E1), ubiquitin-conjugating enzymes (E2), and ubiquitin–protein ligases (E3) are the three vital enzymes in the ubiquitination process that catalyze the attachment of ubiquitin to lysine residues in substrate proteins and promote degradation. E3 ligases cause the association of this ubiquitin with the Protein Of Interest (POI), which acts as a signal for proteolytic enzymes to start breaking down the proteins (Figure 1) [15]. On the basis of this ubiquitin–proteasome system, the PROTAC (proteolysis-targeting chimaeras) technology was developed for target protein degradation [16].

PROTACs are heterobifunctional molecules with the following three structural features: (i) a ligand that binds to the target protein, (ii) a ligand that recruits an E3 ligase, and (iii) a suitable linker that connects these two ligands. This structure allows the POI and E3 ligase to become adjacent and accessible for conjugation as the PROTAC molecule becomes sandwiched between these two proteins. Later, the E3 ligase complex ubiquitinates the POI and leads to its identification and degradation by the 26S proteasome, which is a component of the ubiquitin–proteasome system in eukaryotes. A single PROTAC molecule may carry out this catalytic process repeatedly, permitting several turnovers of ubiquitylation events that result in the degradation of the recruited substrate [17]. 

Three different steps are involved in the ubiquitin–proteasome system. In the initial step, the E1 ubiquitin-activating enzyme uses ATP to activate ubiquitin, resulting in an activated ubiquitin-adenylate. This intermediate is then transformed into a thioester by joining with an E1 enzyme’s catalytic cysteine. In the second step, the activated ubiquitin is transported through a transthioesterification reaction to the catalytic cysteine of an E2 ubiquitin-conjugating enzyme. The carboxy terminus of ubiquitin then forms an isopeptide bond with the side chain of a lysine residue on the surface of the target protein after the formation of a ternary complex between the E2 enzyme, an E3 ubiquitin ligase, and the substrate protein. The ubiquitin protein itself contains seven lysine residues, which provides options for other ubiquitins to bind and initiate polyubiquitination. The fate of the ubiquitinated substrate protein depends on the fashion in which the polyubiquitin chain has formed. Ubiquitin chain forming via linkage between k48/k11 lysine residues of ubiquitin protein usually results in proteolysis. This ubiquitin chain is eventually recognized by the 26s proteasome system, which degrades the substrate protein thereafter [17,18]. 

### 2.1. Types of Targeted Protein Degraders

In addition to PROTACs, there are various modalities that can degrade the targeted protein via proteasomal degradation, such as molecular glues [19], double-mechanism degraders [20], selective AR degraders (SARD) [21], hydrophobic tagging technologies A(HyT) [22], the transcription factor PROTAC (TF-PROTAC) [23], the dual-PROTAC [24], and the selective estrogen receptor degrader (SERD) [25]. Apart from proteasomal degradation, TPD via lysosome is also being employed by researchers. This pathway can go via two ways, namely the endosome–lysosome system and the autophagy–lysosome system. The drugs that target the endosome–lysosome system include the lysosome-targeting chimera (LYTAC) [26], bispecific aptamer chimera [27], antibody-based PROTAC (AbTAC) [28], and GlueTAC [29]. Drugs that target the autophagy–lysosome system include the autophagy-targeting chimera (AUTAC) [30], autophagosome tethering compound (ATTEC) [31], autophagy-targeting chimera (AUTOTAC) [32], and chaperone-mediated autophagy-based degrader (CMA-based degrader) [33,34].

### 2.2. Mechanism of Action of ARV110 in Treating Prostate Cancer

Currently, there are three PROTACs targeting the AR in clinical trials (Table 1). One of them is ARV-110, which was developed by Arvinas. Of all the PROTACs in clinical trials, ARV-110 was the first to enter clinical trial. ARV110 degrades AR by inducing protein degradation via hijacking the above mentioned pathway of UPS. ARV-110 shows its activity by bringing the AR and ubiquitin-recruiting enzyme (CRBN E3 ligase) in to close proximity. This is achieved when the thalidomide warhead and another component of ARV-110 bind to the E3 ligase covalently and AR non-covalently, respectively, forming a ternary complex (E3 ligase-ARV-110-AR). The presence of the E3 ligase and AR makes the process of ubiquitination more efficient. Once the AR is ubiquitinated, it is recognized and broken down by 26s proteasome (Figure 2). The free PROTAC and E3 ligase complex are then ready to form a ternary complex with another AR [3].

## 3. Advantages and Disadvantages of PROTACs over SMIs

PROTAC technology has recently garnered significant interest among researchers in the field of cancer therapeutics, especially in the area of prostate cancer drug discovery. This is due to the fact that PROTACs possess distinct benefits and variations compared to the traditional small molecule inhibitors. (Table 2) [18].

The development of effective anticancer drugs poses challenges in terms of targeting oncoproteins and achieving effective binding with them. However, PROTACs offer a unique solution as they can target proteins without binding to the active site, making previously undruggable targets now druggable [35]. In contrast to PROTACs, conventional small-molecule inhibitors (SMIs) require larger doses to be effective because they need to bind with many target proteins. The unique aspect of PROTACs is that they do not have to bind to the active site of the target protein, making previously undruggable targets now druggable. The “hook effect” occurs when too many PROTAC molecules bind to either the target protein or E3 ligase, reducing the efficiency of the ternary complex formation and leading to less target degradation. This is why low doses of PROTACs are preferred, as a single molecule can form the ternary complex by binding with both the target protein and the E3 ligase. The catalytic nature of PROTACs allows for longer action and a different PK/PD profile compared to SMIs. PROTACs only need a small amount of interaction with the target to induce noticeable degradation. Their event-driven mechanism of action has a wider biological relevance than SMIs as it can overcome resistance mechanisms such as drug target overexpression and resistance-causing mutations.

Another significant advantage is that proteins that can be degraded using PROTAC do not require an enzyme active site but a small-molecule binding site that an E3 ligase can access. A ligand must typically have a moderate affinity to use these sites, and a recruited E3 ligase must have access to the POI surface close to the binding site. High-affinity ligands are generally not required for this process, which is usually strenuous to design. An ideal E3 ligase ligand would be a powerful binder with a delayed off-rate, even though the ligand affinity for the POI might be modest. The capacity to turn over and function repeatedly is lost by PROTACs that covalently bind to the POI. In contrast, PROTACs that covalently bind to an E3 ligase would decrease the three-body assembly kinetics to a two-body issue and increase catalytic efficiency [17,18,36]. Crew’s lab presented examples of successful degradation of three RTKs, EGFR, HER2, and c-Met, including various mutants of EGFR and c-Met. This demonstrates that PROTACs are capable of causing the degradation of active receptor tyrosine kinases.

Like any other technique, PROTAC technology has both advantages and disadvantages. The main issue with PROTACs is that they do not conform to the usual “rule of 5”, unlike traditional, small-molecule drugs [37]. The characteristics and molecular weights of PROTAC can be altered by the introduction of short linkers, but ‘linker-ology’ is more than just a means of joining the POI and E3 warheads. Another significant challenge is the limited availability of ligands for E3 ligases, with only a small number of E3 ligases such as CRBN, VHL, MDM2, IAPs, DCAF15, DCAF16, RNF114, KEAP1, FEM1B, etc., out of the 600 known E3 ligases being utilized in clinical practice [38]. Although research is being conducted in this area, cell surface proteins are currently not considered optimal targets for PROTAC therapy as the UPS resides inside the cell [17]. The choice of the ideal linker to join these two binding components has proven to be a significant design challenge for PROTACs to date. Most of the time, scientists use synthetically accessible chemistry to experiment with linkers of varying lengths. However, the research has shown that sometimes, having a linker of an arbitrary length can negatively impact the target. In other cases, the cooperation between the target and the E3 ligase is dependent on a protein–protein interface that includes the linker. There is an increasing body of evidence that suggests that a stable ternary complex, formed by protein–protein interactions, is crucial for successful degradation [39].

## 4. Genetic Alterations Affecting Prostate Cancer

PC is one of the most heritable human cancers with a strong genetic component. An integrated investigation of advanced prostate cancer found that nearly 90% of individuals with metastatic castration-resistant prostate cancer (mCRPC) had clinically actionable germline and somatic alterations. Both somatic and germline genetic aberrations can affect the tumorigenesis as well as the disease progression of PC [40]. The genetic abnormalities of PC that cause sporadic cancer are predominantly somatic mutations. The most common changes in prostate cells are the activation of oncogenes and the loss of function of tumor-suppressor genes. Additionally, structural lesions such as genomic rearrangements, including amplifications, translocations, or deletions, are a leading cause of the development of PC [41]. Copy number variations, such as the gain or loss of segments of genomic DNA, can also lead to the amplification of oncogenes and deletion of tumor-suppressor genes. Another potential outcome of chromosomal rearrangements is the formation of gene fusions that promote oncogenesis. Single nucleotide polymorphisms (SNPs), point mutations such as missense and nonsense mutations, and frameshifts are less prevalent in prostate cancer [42].

The development of prostate cancer is mediated by abnormalities in AR signaling, and androgen deprivation therapy (ADT) is the cornerstone of systemic treatment for individuals with advanced PC. On the other hand, research findings in the biology of PC have revealed that up to 60% of individuals with advanced PC have clinically actionable molecular and genetic alterations in non-AR-related pathways (Figure 3) [43]. Hence, it is essential to focus on both AR- and non-AR-related targets for improvements in PC therapy (Figure 4). 

### 4.1. Non-AR-Related Pathways

A spectrum of genetic alterations was observed in different molecular pathways of PC, including DNA repair, Phosphoinositide 3-Kinase (PI3K), Ras/Raf/MAPK, TGF-β/SMAD4, epigenetic deregulation, and many more pathways [44]. The incidence of germline mutations in DNA repair genes is significantly higher in metastatic PC as compared to localized PC. Genomic studies have identified germline or somatic aberrations in various DNA repair genes such as ATM, BRCA1, BRCA2, CHEK2, FANCA, RAD51D, and PALB2. BRCA2 was the most frequently altered of the major DNA repair genes, followed by ATM. Deletions (BRCA2, ATM, and PALB2), frameshift mutations (BRCA1 and CHEK2), SNPs (RAD51D), and loss of functional mutations (FANCA) are some of the genetic aberrations observed in PC [45]. Phosphatase and Tensin homologue (PTEN) is a tumor-suppressor gene that negatively regulates the PI3K pathway. PTEN appears to be more often mutated in PC metastases, indicating a function for PTEN in the disease progression. The common alterations in the PTEN gene include deletions and frameshift mutations. Furthermore, the Ras/Raf/MAPK pathway is a critical signaling mechanism that plays a critical role in many cancers. Ras and Raf promote MAPK signaling and may increase AR transcriptional activity. The point mutations in RAS oncogenes (HRAS, KRAS, and NRAS) transform the normal cellular genes into abnormally activated oncogenes, which can lead to tumorigenesis in PC. 

The E26 transformation-specific (ETS) family of transcription factors, such as ETVs and ERG genes, are involved in the regulation of cell cycles, cell proliferation, and several cellular functions. The development of PC is associated with recurrent gene fusions in ERG, ETV1, ETV4, and ETV5, as well as deletions and translocations in ETS2 [46]. SMAD4 serves as a mutual downstream node of the transforming growth factor-beta (TGF-β) and bone morphogenetic protein (BMP) pathways and controls cell proliferation. Deletions in the SMAD4 gene affects the TGF-β/SMAD4 pathway that leads to the metastatic progression of PC [47]. Annexin A7 (ANXA7) gene encodes Ca^2+^-activated GTPase, which promotes cell proliferation. ANXA7 expression was shown to be often lost in prostate cancer, particularly in metastasis and local recurrences of hormone-refractory prostate cancer. In PC, the hemizygous deletion of ANXA7 leads to a decreased expression and function that results in cancer progression [48]. AT motif binding factor 1 (ATBF1) gene expression is linked to reduced cell proliferation, increased expression of the CDKN1A (p21) tumor suppressor, and decreased expression of the AFP oncoprotein. Hence, ATBF1 is an equitable candidate tumor-suppressor gene in PC. A deletion of 21 or 24 nucleotides in the coding region has been found not only in sporadic tumors, but also in the germline DNA of PC [49]. 

The CDKN1B gene provides the instructions for making the p27 protein, which plays a crucial role in controlling cell division. A variation of the CDKN1B gene, which is known as an SNP variant, has been linked to an increased risk of advanced prostate cancer (PC). Additionally, the absence of the CDKN1B gene leads to abnormal cell division [50]. CHEK2, a checkpoint kinase, regulates the p53-mediated DNA damage signaling pathway and has been found to have several alterations, including frameshift mutations, in both hereditary and sporadic prostate tumors [51]. The EPHB2 gene codes for a receptor tyrosine kinase protein and has been found to have both frameshift and nonsense mutations in 10% of sporadic PC cases. 

Studies have investigated whether certain genetic alterations in Glutathione S-transferases (GSTs)—GSTP1, GSTM1, and GSTT1—that play a role in carcinogen metabolism and defense against reactive oxygen species are linked to prostate cancer risk. The results showed that specific mutations and SNPs in GSTP1, GSTM1, and GSTT1 have been linked to an increased risk of prostate cancer [52]. The Kruppel-Like Factor 5 (KLF5) is a transcription factor that is involved in cell proliferation, differentiation, and carcinogenesis. The hemizygous and homozygous deletion of KLF5 gene in human PC is rare [53]. Further, KLF6 is a zinc finger transcription factor that has been linked to cell proliferation and differentiation. A germline SNP in KLF6 is significantly associated with an increased relative risk of PC [54]. 

Macrophage scavenger receptors (MSRs) are trimeric membrane glycoproteins that facilitate the binding, internalization, and processing of a wide array of macromolecules. Genetic aberrations in MSR1, including truncating mutations, have been presented to be associated with the PC risk in both sporadic and hereditary cancers [55]. MYC is a cellular proto-oncogene that is well associated with cell transformation. The overexpression of MYC can immortalize prostatic epithelial cells. Hence, the gain of functional mutations identified in MYC is clearly an oncogenic factor in PC [56]. NK3 Homeobox 1 (NKX3-1) is a prostate-specific gene in humans. The NKX3-1 gene is expressed at higher levels in mature prostate tissues, but its expression is decreased in prostate cancer cells. Due to the gene being haploinsufficient, a hemizygous deletion leads to a reduction in both its expression and function [57]. TP53 is a tumor-suppressor gene whose modifications are implicated in the molecular genetics of several human cancers. Furthermore, the p53 gene is one of the most commonly mutated genes in human cancers. While mutations of p53 are relatively rare in early-stage prostate cancer, they become more frequent in higher-stage, higher-grade tumors and metastases. Deletions and functional mutations of the p53 gene have been found to be prevalent in advanced prostate cancer [58].

The human serum paraoxonase and arylesterase 1 (PON1) gene is involved in the elimination of carcinogenic lipid-soluble radicals. The expression of PON1 varies widely in humans, and specific polymorphisms in the gene have been linked to different levels of PON1 protein in the blood. Single nucleotide polymorphisms (SNPs) in the coding region of PON1 have been associated with a decreased serum paraoxonase activity, which may contribute to the development of cancer [59]. The RNASEL gene encodes a 2′-5′-oligoadenylate (2–5 A)-dependent RNase L protein that plays a vital role in the innate immune response. The variant (Arg462Gln) in RNASEL is allied with fewer enzymatic activities and was significantly associated with PC risk. The frameshift, truncating mutation, as well as the deletions in this gene are linked to an elevated risk of PC [60]. Steroid 5-alpha-reductase 2 (SRD5A2) converts testosterone to di-hydro testosterone, which is essential for the development of the prostate. The SNP variants in SRD5A2 are associated with more aggressive features of PC such as a higher tumor stage, metastasis, and extracapsular disease [61]. In addition, prostate cells express a vitamin D receptor (VDR), which mediates the functions of 1, 25-dihydroxy vitamin D, and the polymorphisms in VDR are connected with an increased risk of more aggressive PC [62]. 

### 4.2. AR Pathway

Androgens play a crucial role in the growth and maintenance of the prostate. The relationship between androgen levels and prostate cancer (PC) has been well established. The intracellular AR, a member of the ligand-dependent transcription factor family, is responsible for mediating androgen activity. When testosterone binds with AR, it triggers the transcription of androgen-responsive genes, thus controlling the growth of both the normal prostate gland and PC [63]. The androgen-signaling axis is a primary target for the treatment of metastatic prostate cancer through androgen ablation therapy. However, this treatment method can become ineffective over time as a result of resistance. Recent findings suggest that this resistance is caused not by a lack of androgen signaling but by genetic alterations that lead to abnormal activation of the androgen-signaling axis. It has been observed that many primary and metastatic prostate cancers exhibit genetic changes in the androgen signaling pathway, such as amplifications, mutations, and changes in AR corepressors (NCOR1/2) and coactivators (NCOA1/2), which result in castration resistance [64]. 

Further, structural rearrangements of AR were identified in one-third of metastatic PCs. In general, FOXA1 suppresses the androgen signaling pathway and promotes prostate tumor growth. Therefore, recurrent mutations in the AR associating factor FOXA1 lead to the development of PC and the alterations in FOXA1 have been identified in 3–4% of both untreated, localized PC and metastatic PC [47].

A significant number of prostate cancer patients have mutations in specific genes, with these alterations being more common in metastatic cases compared to localized ones. Traditional small-molecule inhibitors (SMIs) struggle to target these mutations, as multiple genetic abnormalities lead to resistance in prostate cancer. Each mutation in a particular gene can alter protein signals differently, so it is important to carefully select the correct gene to target. Some genetic abnormalities, such as SNP, deletions, and point mutations, can render the target protein untreatable. To address these mutations, PROTACs (Proteolysis-Targeting Chimera) offer a promising alternative therapeutic option for prostate cancer. Currently, PROTACs are designed to target the AR in prostate cancer. While other genes in addition to AR are altered in prostate cancer, AR remains the primary focus of established PROTACs. These alterations and targets could be valuable for the development of new PROTACs for precision cancer therapy and medicine for prostate cancer.

## 5. Prominent PROTACs Targeting Prostate Cancer

Over the past several decades, the pathophysiology of prostate cancer has been extensively studied and understood. Inhibiting the AR is widely considered as one of the most effective therapies for prostate cancer, despite its tendency to develop resistance over time. To address this issue, alternative approaches have been proposed, including the use of PROTACs, which target and degrade the AR. The removal of the protein reduces the risk of developing resistance and has gained significant attention in the treatment of prostate cancer, particularly after the development of ARV-110. A significant number of PROTACs (nearly 170) have been reported to target various proteins that are associated with prostate cancer (Appendix A). In this article, we focus on the development of various AR and non-AR target degrading PROTACs and their potential to treat prostate cancer, with an emphasis on drug optimization and medicinal chemistry.

### 5.1. Androgen Receptor Targeting PROTACs

#### 5.1.1. CRBN-Based PROTACs

There are now a significant number of unique AR degraders that have been revealed and are based on the target protein degradation (Proteolysis-Targeting Chimera) principle. Lukas et al. synthesized three series of degraders based on FDA approved AR inhibitor, enzalutamide. They modified the exit vector portion of the enzalutamide with benzamide, phenoxy, and piperidine scaffolds along with linker composition. The subseries with an exit vector of biphenyl/phenoxy type, demonstrated effective AR degradation at a concentration of 1 µM. A polyethylene glycol (PEG) linker containing PROTAC was discovered to be active compared to simple alkyl chain. The final subseries had PROTACs with a phenoxy exit vector with more rigid linkers, which resulted in the best molecules. Furthermore, stereochemistry of the molecules influenced the target degradation, as the geometrical orientation of the ternary complexes was likely altered. PROTAC 35 (compound **1**) from final subseries was chosen for additional in-depth investigation since it significantly degraded even at 0.1 mM, resulting in DC_50_ values of 75 nM and 77 nM in A549 and LNCaP cell lines, respectively. Additionally, they carried out cell specificity tests, which demonstrated that compound **1** was more selective for the PC cell line LNCaP [65].

A series of CRBN-mediated AR PROTACs, which combine a tetramethylcyclobutane-based AR antagonist with the TD-106 CRBN binder, were developed and evaluated by Akshay D. Takwale et al. for the treatment of metastatic castration-resistant prostate cancer. Among the synthesized, TD-802 (compound **2**) displayed the most effective AR protein degradation, with a DC_50_ of 12.5 nM and a maximum degradation of 93% in the LNCaP cell lines. The effectiveness of AR degradation is highly dependent on the linker location within the TD-106 (CRBN binder). In addition, pharmacokinetic studies in mice demonstrated that over 70% of the degraders, including TD-802, exhibited good microsomal stability after 30 min of incubation, a long half-life of over 4 h, and high plasma exposure following intravenous and intraperitoneal injections. Finally, TD-802 induced AR degradation and demonstrated strong antitumor activity in an in vivo xenograft mouse model, indicating that CRBN-based AR degraders may have potential for use in the development of new AR therapies [66]. 

Jian-Jia Liang et al. developed and tested a series of new PROTACs based on RU59063 derivatives and cereblon ligands for their ability to degrade the AR protein in LNCaP (AR+) cells. Almost all the PROTACs displayed a good binding affinity and efficacy against AR. Among the compounds, A16 (compound **3**) was found to be the most promising, with an 85% AR antagonist efficacy compared to enzalutamide (89%). Further, the Western blot analysis revealed that A16 caused only a minor decrease in the amount of AR protein in LNCaP cells. Molecular docking experiments with AR and DDB1-CRBN E3 ubiquitin ligase complex (PDB: 2AXA) indicated that A16 has binding properties similar to other non-steroidal AR ligands, with key interactions [67].

Ga Yeong Kim et al. designed, synthesized, and evaluated a series of PROTACs consisting of PEG chain-linked bicalutamide analogues and thalidomides (E3 ligase recruiters) as novel AR degraders. Among the synthesized, 13b (compound **4**) was found to be highly effective in decreasing the mRNA levels of transcriptional genes that are dependent on AR, such as prostate-specific antigens and TMPRSS2. It was also more effective than 13c (compound **5**), which had a slightly greater AR inhibitory activity compared to bicalutamide. The author’s findings suggested that replacing the trifluoromethyl and sulfide groups with iodine and ether, respectively, may improve the AR antagonist activity of bicalutamide. Further, the AR antagonist activity was not affected when the hydroxyl group was replaced with fluorine in the terminal phenyl moiety at the solvent exposure site, indicating that the ether linkage may be used as the linker moiety for conjugating the AR ligand with thalidomide. The compounds **13b** and **13c** (compounds **4** and **5**) were significantly degraded by the AR in LNCaP cells in a dose- and time-dependent manner. Interestingly, the AR variant 7 (AR-V) form was more susceptible to compound **13c** than the wild type. These results suggest that compound **13c** targets and degrades AR through direct binding and does not modulate the mRNA expression [68].

In another study, researchers led by Shaomeng Wang reported a highly potent and effective AR degrader ARD-61 (compound **14**) for the treatment of PC. However, it was found to be inactive when administered orally in mice. In order to improve the oral bioavailability of ARD-61, the authors opted to utilize the cereblon as the E3 ligase ligand instead of the VHL ligand, owing to its lower molecular weight and superior drug-like properties. The first series of molecules was synthesized using the AR antagonist of ARD-61 and thalidomide with both flexible and semi-rigid linkers, and it was observed that constrained linkers had almost the same potency as linear and flexible linkers. The next series of molecules was synthesized by changing the linking position from the meta to the ortho position in the phenyl ring of the cereblon ligand, which decreased the activity. The most promising molecule from this series was then tested using modified AR antagonists with the same linker and CRBN ligand, showing that the AR antagonist of ARD-61 is essential for its activity. The DC_50_ values of the promising compounds were tested on the VCaP cell line, resulting in <0.5 nM activity and <5 nM for DC_90_. The potencies of these compounds were also evaluated, and the mechanism of action was tested. Further pharmacokinetic properties showed that the selected compounds have an excellent oral bioavailability. The most promising molecule, ARD-2128 (compound **6**), was tested further and found to be effective in reducing the levels of AR, PSA, TMPRSS2, and FKBP5. Additionally, it was discovered that ARD-2128 is highly effective at reducing tumors, even at lower doses compared to enzalutamide [69].

Weiguo Xiang et al. reported the development of a potent and orally active compound for AR degradation. The initial series was synthesized using AR antagonists from Pfizer and Arvinas, along with the CRBN ligand and linkers of various lengths. The first series of compounds was synthesized by hooking Pfizer’s molecule to the isoindoline-1,3-dione moiety of thalidomide at the five position through the para-position of its phenyl group, with amine groups on both ends and linear alkyl groups of various lengths. Although all the compounds showed promising results similar to ARV-110, with a DC_50_ of 1.6 nM and Dmax of 98% in the VCaP cell line, their oral bioavailability was not ideal. Additional compounds were synthesized using semi-rigid and rigid linkers. Rigid linkers resulted in significant improvements in the DC_50_, Dmax, and kinetic properties. The lead compound was further optimized by modifying the AR antagonist portion of the molecule, specifically by altering the cyclohexane ring and the linker atom connecting the phenyl and cyclohexane rings. The results demonstrated that the cyclohexane ring and tertiary nitrogen (as the linker atom) are highly effective and essential for the activity. The modified compounds using different rigid linkers were identified as highly effective toward AR, with DC_50_ values ranging from 0.01 to 1.4 nM and Dmax values of 93 to 100%. After subjecting the selected AR degraders in LNCaP cell lines to further analysis, including AR degradation kinetics, pharmacodynamics, and tissue distribution studies, it was determined that the most promising molecule among them was ARD-2585 (compound **7**) (Figure 5) [70].

#### 5.1.2. VHL-Based PROTACs

Xin Han et al. analyzed the correlation between the binding affinity of the VHL ligand component of the VHL protein and the efficacy of the resultant PROTAC degraders. The study revealed that utilizing VHL ligands with a low binding affinity (Ki = 2–3 μM) to the VHL protein can result in the development of highly effective and potent degraders. According to prior research, it was observed that co-crystal structures of VHL ligands in conjunction with VHL protein demonstrated that the 4-methylthiazole group of the ligand binds to the hydrophobic region of the VHL protein. Hence, the region was modified with different groups such as nitrile, bromine, chlorine, fluorine, and hydrogen, and the resulting compounds were tested for their binding affinity to VHL protein. Using ARD-61 (compound **14**) as a template, a series of new PROTAC AR degraders was synthesized using VHL ligands with a wide range of binding affinities to VHL. It was found that almost all of the synthesized molecules were effective at reducing AR at all tested concentrations, with three compounds showing excellent activity (greater than 80% degradation at 10 nM), despite having a weak binding affinity to VHL. To further optimize the compound with a very weak VHL binding affinity ARD-266 (compound **8**), the linker was modified by reducing the number of rings and atoms within the rings, resulting in a more potent compound (>90% degradation at 10 nM). The biological evaluation of ARD-266 demonstrated that it has DC_50_ values of 0.5, 1, and 0.2 nM in the LNCaP, VCaP, and 22Rv1 cell lines, respectively. After a 6-h treatment, ARD-266 was able to reduce the AR protein level by more than 95% at concentrations of 30 nM in both the LNCaP and VCaP cell lines and 10 nM in the 22Rv1 cell line. In comparison to conventional AR inhibitors, ARD-266 was found to be more potent, with an IC_50_ value of 6 nM compared to 182 nM for the AR antagonist [71].

Linrong Chen et al. reported a series of AR degraders, including A031, based on two distinct AR antagonists (azabicyclic and fused bicyclic compounds) and four E3 ligands. They developed PROTACs with a variety of connecting options, such as various warheads, and modified the linker using different heterocyclic and phenyl ring combinations along with an alkyl chain, resulting in a moderate increase in activity. The modifications made to the attachment points on the CRBN ligand did not yield any enhancement in activity. Nevertheless, modifications made to the AR antagonist component led to a reduction in activity. It is noteworthy that substituting the E3 ligase ligand with VHL-2 and shortening the linker length led to the development of improved molecules such as A031 (compound **9**). The compound A031 showed promising activity even at lower concentrations, as low as 0.125 µM. Further, the additional biological studies revealed that A031 has comparable efficacy and five times less toxicity as compared to EZLA (the positive control) [72].

Munoz et al. reported a series of niclosamide-based PROTACs for the treatment of AR+ve PC that use VHL-032 as an E3 ligase-binding warhead. The compounds demonstrated moderate IC_50_ values against the PC-3, LNCaP, and 22Rv1 cell lines. However, it was found that their mechanism of action did not involve AR degradation, as confirmed by Western blotting assays, even at concentrations as high as 1 µM. This is in contrast to the clinical candidate ARV-110, which demonstrated AR degradation. However, Niclo-Click PROTAC 5 (compound **10**) exhibited a decent IC_50_ of 1.04 ± 0.18 µM in the LNCaP cell line [73].

Lee et al. designed and developed a unique PROTAC using 4-(4-phenylthiazol-2-yl)morpholine called MTX-23 (compound **11**) which selectively targets and degrades both the full-length AR (AR-FL) and the AR splice variant 7 (AR-V7). In immunoblotting studies, MTX-23 had a DC_50_ of 0.37 µM for AR-V7 and 2 µM for AR-FL. The compound was found to inhibit proliferation and enhance apoptosis selectively in androgen-responsive PC cells. MTX-23 was tested in cells resistant to four FDA-approved SAT agents (abiraterone, enzalutamide, apalutamide, and darolutamide) and exhibited promising results. It decreased cellular proliferation and reduced tumor growth in both in vitro and in vivo, potentially making it effective against castration-resistant prostate cancer by degrading both AR-V7 and AR-FL [74].

Jemilat Salami et al. synthesized a series of PROTACs based on enzalutamide to target the AR using VHL as an E3 ligase-binding warhead. Among the compounds synthesized, ARCC-4 (compound **12**) was the most potent, demonstrating a Dmax of 98% and a DC_50_ of 5 nM. In cellular models of castration-resistant prostate cancer (CRPC), ARCC-4 was more effective at inducing apoptosis and inhibiting proliferation of AR-amplified prostate cancer cells compared to its parent compound, enzalutamide. Additionally, ARCC-4 retained its ability to degrade AR and inhibit cell proliferation in a high-androgen environment. The mechanism of action for ARCC-4 was determined to be proteasomal degradation through polyubiquitination of AR, as confirmed using the Tandem Ubiquitin-Binding Element (TUBE1) pull-down assay. ARCC-4 was also able to effectively degrade several clinically relevant mutants of AR, including AR-F876L, AR-T877A, AR-L702H, AR-H874Y, and AR-M896V. ARCC-4 was effective at degrading the AR protein in VCaP cells even with increasing concentrations of the synthetic androgen R1881 up to 1 nM [75]. 

Xin Han et al. conducted a comprehensive optimization process to develop a highly potent and effective AR-degrading PROTAC. They utilized enzalutamide as the AR-binding ligand and VHL as the E3 ligand and synthesized a set of PROTACs through the use of an amide linker with varying chain lengths. The biological studies of synthesized compounds indicated that an optimal linker length of 11–12 atoms resulted in the most effective AR degradation. The introduction of a pyridine group directly connected to the ethynyl group significantly increased the activity and solubility of the compound. The optimization data demonstrated that increasing the rigidity of the linker enhances the activity of the PROTAC. The researchers then modified the stereochemistry of VHL to examine its role in binding to the VHL protein, eventually selecting ARD-69 (compound **13**) for further investigation. ARD-69 achieved DC_50_ values of 0.86 and 0.76 nM in LNCaP and VCaP cell lines, respectively, and induced greater than 95% AR degradation at 10 nM in both cell lines [76].

Steven Kregel et al. reported the development of the AR degrader ARD-61 (compound **14**), which demonstrated effectiveness in addressing resistance mechanisms that often arise during PC treatment. From the results, it was identified that ARD-61 caused PARP cleavage in all AR-dependent cell lines, regardless of AR levels, AR splice variant expression, or AR mutational status. ARD-61 was also effective in model systems resistant to enzalutamide and AR splice variants, including AR-V7. Although ARD-61 is unable to bind and degrade AR-V7, it was able to inhibit tumor cell growth in models overexpressing AR-V7. In addition, ARD-61 was degraded to full-length AR in all four CRISPR-induced clones overexpressing AR-V7 but had no effect on the production of AR-V7. When tested in the aggressive, metastatic, castration- and enzalutamide-resistant CWR-R1 EnzR xenograft model, which expresses high levels of AR-V7, ARD-61 was found to be very effective in vivo. These findings suggest that, despite the potential for AR-V7 to enhance AR signaling, full-length AR is necessary for the development and survival of castration- and enzalutamide-resistant prostate cancer (Figure 6) [77].

#### 5.1.3. Miscellaneous

Bohan Ma et al. developed a peptide-based PROTAC that utilizes ultra-small gold nanoparticles as a delivery method. The synthesized degraders target the DNA-binding domain (DBD) of AR and induce the degradation of both AR and AR-V7. The desired PROTAC sequence was synthesized using a solid phase peptide synthesis, with a flexible linker introduced between the AR-DBD- and MDM2-targeting sequences. Using isothermal titration calorimetry, the binding affinity between the AR pep-PROTAC and AR-V7 was determined to be 12.2 nM and 49.6 nM, respectively. The Au-AR pep-PROTAC was found to stimulate the degradation of AR and AR-V7 and reduce the growth of AR- and AR-V7-positive prostate cancer cell lines, including LNCaP, C4-2, and CWR22Rv1, with IC_50_ values of 230.8 nM, 248.1 nM, and 126.9 nM, respectively. The DC_50_ values for half-maximal degradation were found to be 175.2 nM, 193.0 nM, and 48.8 nM (AR)/79.2 nM (AR-V7) in LNCaP, C4-2, and CWR22rv1 cells, respectively. Furthermore, it was determined that the Au-AR pep-PROTAC compound elicits degradation of AR in both the cytoplasmic and nuclear compartments [78].

Hang Xie et al. reported the synthesis of two series of selective AR degraders (SARDs) based on the AR agonist RU59063 (4-(4,4-dimethyl-3-(4-hydroxybutyl)-5-oxo-2-thioxo-1-imidazolidinyl)-2-(trifluoromethyl)benzonitrile) with hydrophobic degrons. The analysis of the structure–activity relationship revealed that compounds with alkyl chain linkers had stronger antiproliferative activity compared to those with phenyl groups. The presence of the 1,2,3-triazole group also significantly impacted the compounds’ activities. A9 (compound **15**) was found to have a strong inhibitory effect against LNCaP cells and a shared AR-binding affinity with enzalutamide. The molecular docking study revealed that the interaction between the extensive hydrophobic tagging chains of the synthesized compounds and the AR could affect the rate of degradation (Figure 7) [79].

### 5.2. Non-AR-Targeting PROTACs

There are a variety of proteins that have been identified as potential targets for the treatment of prostate cancer. Apart from the AR, PROTACs have been developed to target some of these proteins. For example, c-Myc and protein kinases, which are frequently overexpressed in PC and have been linked to cancer progression. These non-AR-targeting PROTACs have the potential to be used as therapeutic agents for the treatment of PC.

Kanak Raina et al. reported the use of PROTAC-induced BET protein degradation for the treatment of castration-resistant prostate cancer (CRPC). They demonstrated that the small-molecule pan-BET degrader ARV-771 (compound **16**) was more effective than BET inhibition in cellular models of CRPC. In order to optimize the lead compound for in vivo studies, the linker’s length and composition were modified to attain the desired pharmacological properties. In vitro studies on the CRPC cell lines 22Rv1, VCaP, and LnCaP95 showed that ARV-771 potently degrades BRD2/3/4 with a DC_50_ smaller than 5 nM. The loss of the BET function was determined by measuring the concentration of the c-MYC protein, which decreased to an IC_50_ smaller than 1 nM. The induction of apoptosis by ARV-771 was demonstrated by the substantial breakage of poly (ADP-ribose) polymerase (PARP) in 22Rv1 cells. ELISA and immunoblotting studies exhibited that ARV-771 dramatically reduced the AR protein levels in VCaP cells, but not JQ1 or OTX015. When the VCaP cells were given a 10 nM dose of ARV-771, the levels of both the FL-AR and AR-V7 mRNA decreased. The in vivo studies of ARV-771, administered at a dose of 10 mg/kg, in male Nu/Nu mice with AR-V7+ 22Rv1 tumor xenografts, demonstrated a significant reduction in the AR-V7 levels in the tumors [80].

Rong Hu et al. designed and synthesized the selective BRD4-PROTAC (WWL0245) using a dual BET/PLK1 inhibitor, WNY0824. Computational analyses revealed that a linker and an E3 ligand, such as lenalidomide and thalidomide, could be introduced to the 1-methylpiperidin-4-yl motif of WNY0824, which is exposed to the solvent region. WWL0245 (compound **17**) showed selective cytotoxicity in BETi-sensitive cancer cell lines, including AR-positive PC cell lines. The authors evaluated the effect of WWL0245 and WWL0202 (compound **18**) on the expression of BRD2/3/4, PLK1, and c-Myc in VCaP cells. Both compounds significantly reduced the BRD4 levels and inhibited c-Myc at concentrations as low as 10 nM but had no significant impact on protein levels. Due to its exceptional antiproliferative activity, selectivity, and low molecular weight, WWL0202 was selected for further studies. WWL0202 was found to have a DC_50_ in the sub-nanomolar range, and when tested on 22Rv1 and VCaP cells, it resulted in a reduction in the BRD4 and c-Myc levels in a concentration-dependent manner [81]. 

Fei Zhou et al. designed multiple cyclin-dependent kinase (CDK)-targeting PROTAC degraders. Using computational studies, they identified the hydrophobic region of two pan-CDK inhibitors (AT-7519 and FN-1501) as the linker attachment points. They designed two series of protein degraders by connecting the two ligands and the CRBN ligand with the same set of linkers. The degraders made with AT-7519 selectively degraded CDK2 at a concentration of 1 µM. After performing a CCK8 assay against PC-3 cell lines, A9 and F3 (compounds **19** and **20**) were found to be potent with IC_50_ values of 0.84 and 0.12 µM, respectively. Mechanistic studies revealed that it could effectively and selectively reduce the levels of CDK2 and CDK9 in a concentration-dependent manner, with DC_50_ values of 62 and 33 nM, respectively. Further synthesis of molecules based on F3 did not result in any improved activity. The compound F3 was found to significantly extend the S phase by 9.8% at a concentration of 250 nM, and higher concentrations caused cell cycle arrest at the G2/M phase. These findings provide a strong platform for targeting non-AR targets for PC therapy using the TPD (targeted protein degradation) strategy (Figure 7) [82].

## 6. Design and Medicinal Chemistry of PROTACs

Prior to 2008, PROTACs were primarily peptide-based; however, the introduction of the first small-molecule PROTAC by the Crews group, utilizing MDM2, marked a pivotal shift in their synthesis. The crucial part of developing PROTACs is the rational design of all the three components: (i) the E3 ligase warhead (E3 ligand), (ii) the target-binding warhead, and (iii) the linker [83].

### 6.1. Design of E3 Ligands

To date, researchers have identified more than 600 E3 ubiquitin ligases in the human genome; however, the potential of these ligases remains constrained by the dearth of small molecules exhibiting a strong affinity and specificity for them [84]. When designing a E3 ligase ligand for specific protein degradation, there are a few important factors to consider, such as the expression levels of different types of E3 ligases in the tissue where the target protein is concentrated, and the E3 ligase selectivity for the target protein. However, for any given target protein, VHL and CRBN ligands are a good starting point due to their wide scope of efficacy [85,86]. Based on the existing VHL and CRBN ligands, many other analogues that mimic their activity were reported using modern techniques, such as fragment-based drug design, structure-based drug design, high-throughput screening, DNA-encoded libraries, phage display technology, etc., and have been employed in the design of novel PROTACs [87].

### 6.2. Design of Linker

The linker plays a critical role in the formation of ternary complexes in PROTAC design, with various elements of the linker involved in specific interactions. As a result, optimizing the linker is a key goal in the development of PROTACs, considering factors such as length, composition, stiffness, and attachment points [88]. The important points to be considered when designing linkers are linker length composition and the linker’s ability to target protein the attachment site (the solvent-exposed site of the ligand). The linker length, if not appropriate, can affect the proximity between E3 ligase and target proteins, which subsequently affects activity. Tertiary structure formation between E3 ligase-PROTAC-target protein, pharmacokinetic properties of PROTAC depends on linker composition [86]. For example, the rigidity of the linker significantly increases the kinetic and dynamic properties of PROTACs [70]. 

### 6.3. Design of a Target Protein-Binding Ligand

Most of the ligands used to design this component of PROTACs are knowledge-based. They usually use ligands that are already approved as inhibitors of POI or reported to have high binding affinity with the target protein. It is not necessary to consider the active binding site of the protein when designing a ligand, unlike conventional inhibitors. It can bind to any site of the protein unless it is not inducing degradation [86]. 

## 7. Conclusions and Future Recommendations

AR plays a key role in the development and progression of prostate cancer, and genetic alterations in the AR gene have been identified in a significant proportion of PC cases. These genetic changes can result in different variants of the AR protein, which can affect its function and response to treatment. This diversity of AR variants can make it difficult to develop drugs that effectively target the receptor, as a single therapy may not be able to inhibit all of the different variants. This is a major challenge in the development of small-molecule therapeutics for PC to target all the AR aberrations. However, this theory has been repeatedly refuted by targeted protein degradation. It has been 22 years since the advent of the targeted protein degradation strategy utilizing PROTACs as a means of inducing protein degradation [89]. The clinical translation of PROTACs has been demonstrated through the initiation of several clinical trials, indicating their therapeutic potential. Furthermore, a plethora of PROTACs targeting both AR and non-AR proteins have been described in the literature, with a specific focus on PC. The ability of certain PROTACs to effectively degrade these AR variants highlights their potential as a promising strategy for overcoming treatment resistance in prostate cancer and other diseases. 

Although most PROTACs developed for PC have focused on the AR, it should be noted that non-AR proteins also contribute significantly to the pathogenesis of PC, comprising 60% of the disease. Despite the significance of non-AR proteins, no PROTACs have been developed to specifically target these proteins. These non-AR proteins often exhibit a variety of genetic alterations, making them difficult to target with small-molecule inhibitors. In the current review article, we summarized the genetic alterations in non-AR proteins and discussed how targeting the same through PROTACs could be an effective strategy in the treatment of PC, as they may overcome resistance mechanisms that are associated with traditional targeted therapies. Further, the use of PROTACs as a precision medicine approach for the treatment of PC is a promising area of research. PROTACs have the potential to selectively degrade specific proteins that drive the disease, thereby offering a more targeted and personalized approach to the treatment of prostate cancer for each individual patient. This method of therapy has the potential to improve the efficacy and specificity of treatment, leading to better patient outcomes. In addition, the use of PROTACs can also lead to the identification of new therapeutic targets that may have been previously undiscovered. The potential of PROTACs as a precision medicine approach for prostate cancer treatment warrants further investigation in preclinical and clinical studies.

The field of PROTAC design and synthesis is challenged by the need for further exploration in the development of ternary complex co-crystal structures using silico methods, the identification of additional E3 ligase ligands, the optimization of synthetic procedures, and the enhancement of PROTAC pharmacokinetic properties. Additionally, there is a need to optimize the binding affinity and specificity of ligands and to gain a deeper understanding of the PROTAC mechanism of action. 

Furthermore, in silico approaches such as Rosetta, PRosettaC, PROTAC-Model, and others can also aid in rationalizing PROTAC-mediated ternary complex formation and enhance PROTAC development efficiency. The adoption of these technologies in the design, development, and optimization of PROTACs may prove to be a promising and time-efficient method in the field of drug discovery. In addition, novel strategies, such as molecular glue, LYTAC, AUTAC, ATTEC, and RIBOTAC, which also promote degradation depending on proximity, are broadening the opportunities for drug development and may have therapeutic advantages [19,26,30,31,90].

The role of artificial intelligence in the design and development of target protein degraders (PROTACs) is a rapidly growing area of research that has enormous potential. By incorporating AI techniques, such as machine learning, computer-aided drug design, and molecular dynamics simulations, the discovery of highly specific and potent PROTACs can be accelerated. A recent study published in the Journal of Chemical Information and Modeling utilized deep learning algorithms to identify new E3 ubiquitin ligase-based PROTACs, showing improved binding affinity and specificity compared to traditional designs [91].

Another study published in the Journal of Medicinal Chemistry applied a machine learning approach to optimize the linker component of PROTACs, leading to improved stability and specificity [92]. These findings demonstrate the potential for AI to significantly enhance the efficiency and success rate of PROTAC design and development. It is imperative that further research be conducted in this field to fully leverage the benefits of AI and contribute to the advancement of therapeutic solutions for various diseases. As the use of AI continues to grow in the field of medicinal chemistry, it is expected to play an increasingly important role in the design and development of PROTACs. The integration of AI into PROTAC research has the potential to revolutionize the way we approach drug discovery and development, leading to faster and more effective solutions for various diseases.

## Figures and Tables

**Figure 1 molecules-28-03698-f001:**
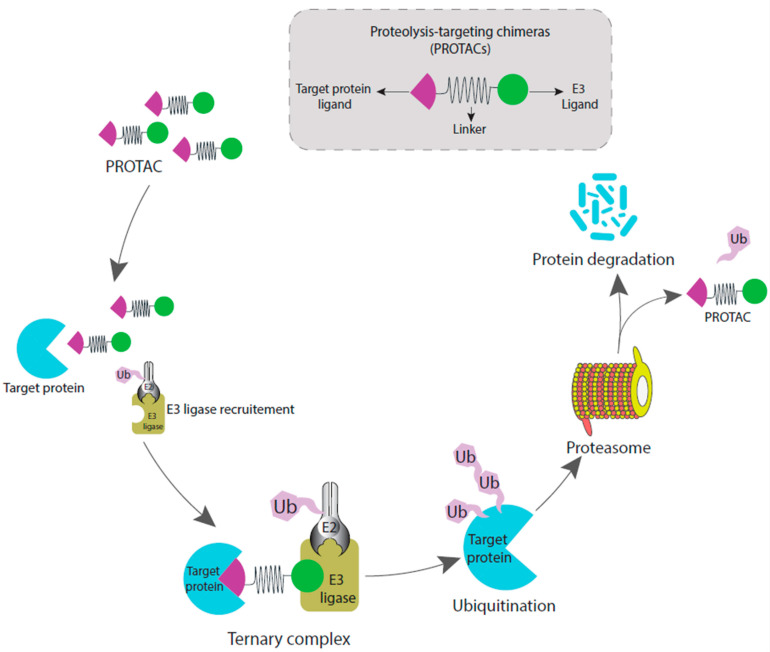
General mechanism of PROTAC-mediated ubiquitination and proteasomal degradation of target protein. The heterobifunctional PROTAC is composed of a ligand that binds to the target protein and a ligand that binds to the E3 ubiquitin ligase by a linker.

**Figure 2 molecules-28-03698-f002:**
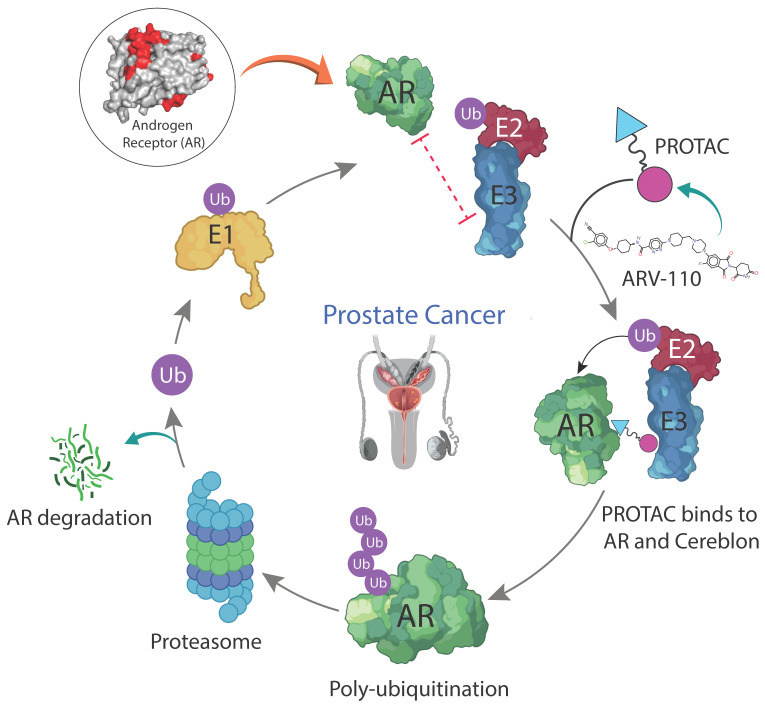
Mechanism of ARV110 in the degradation of AR protein. Initially, the ubiquitin protein binds to an E1 ubiquitin-activating enzyme and then transfers it to an E2 ubiquitin-conjugating enzyme. ARV110 binds with cereblon E3 ligase, AR, and forms a ternary complex. Finally, the cereblon E3 ligase delivers ubiquitin to the AR and induces poly-ubiquitination. The ubiquitinated AR proteins are recognized and degraded by proteasome.

**Figure 3 molecules-28-03698-f003:**
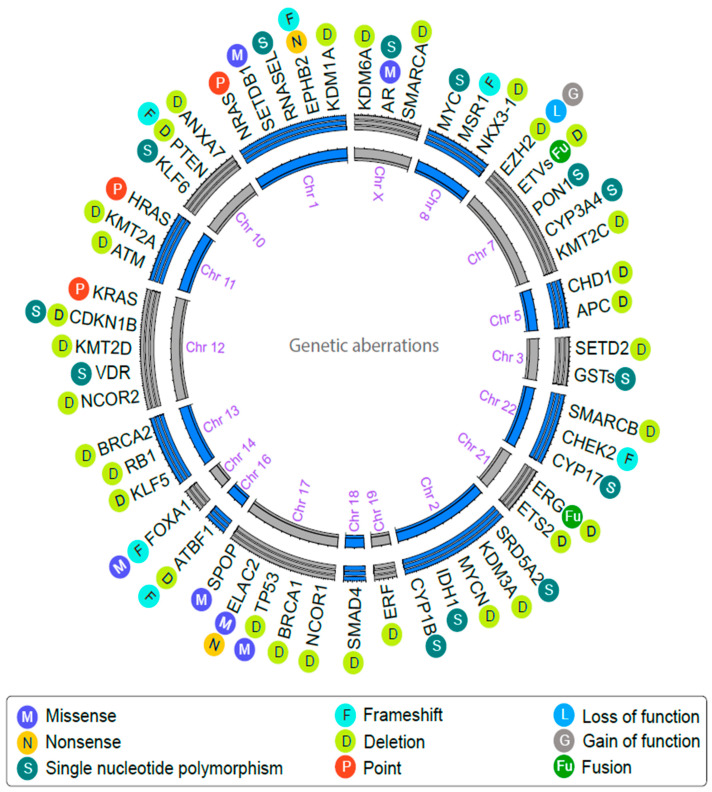
Overview of genetic aberrations in different genes involved in the progression of prostate cancer. Genes and their respective chromosomal location were depicted in outer and inner circles, respectively. Each genetic variation is denoted with a single letter along with a distinct color.

**Figure 4 molecules-28-03698-f004:**
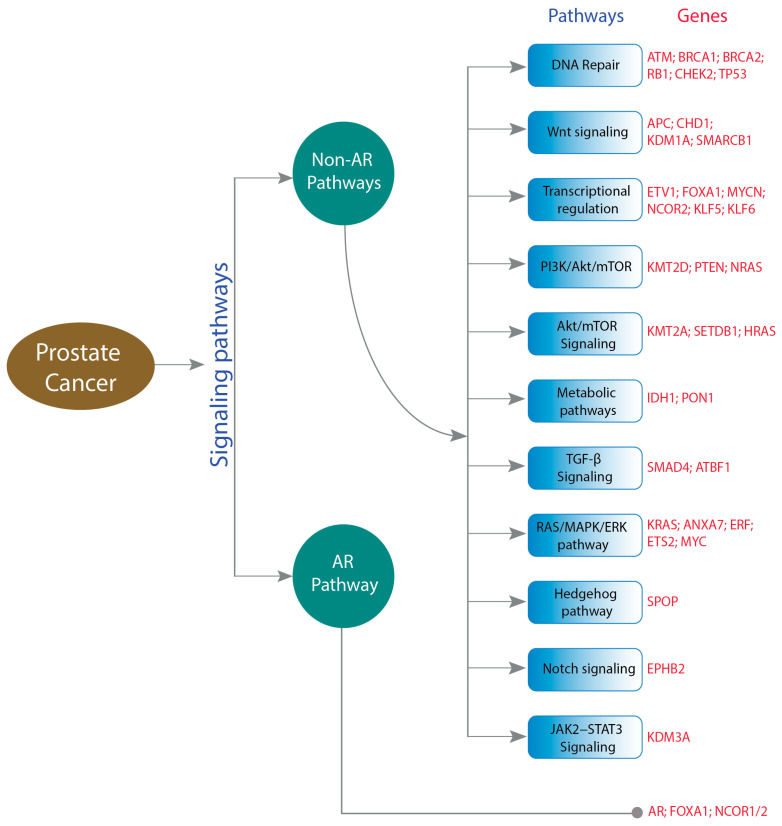
The intricate signaling pathways, including both androgen receptor (AR) and non-AR pathways, and their respective genes have been implicated in the progression of prostate cancer. The relevant genes have been highlighted in red.

**Figure 5 molecules-28-03698-f005:**
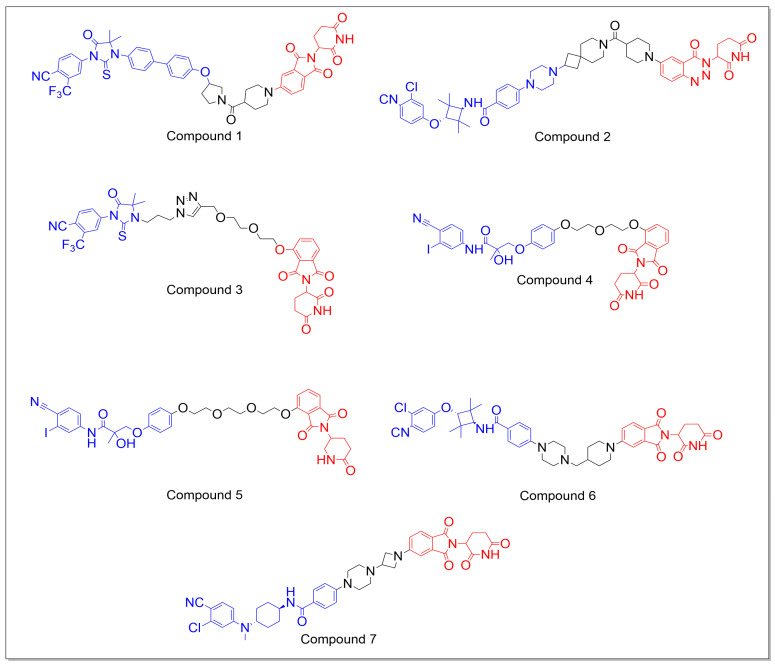
The chemical structures of CRBN-based PROTACs that target the androgen receptor typically display the POI ligand in blue, the E3 ligase ligand in red, and the linker region in black.

**Figure 6 molecules-28-03698-f006:**
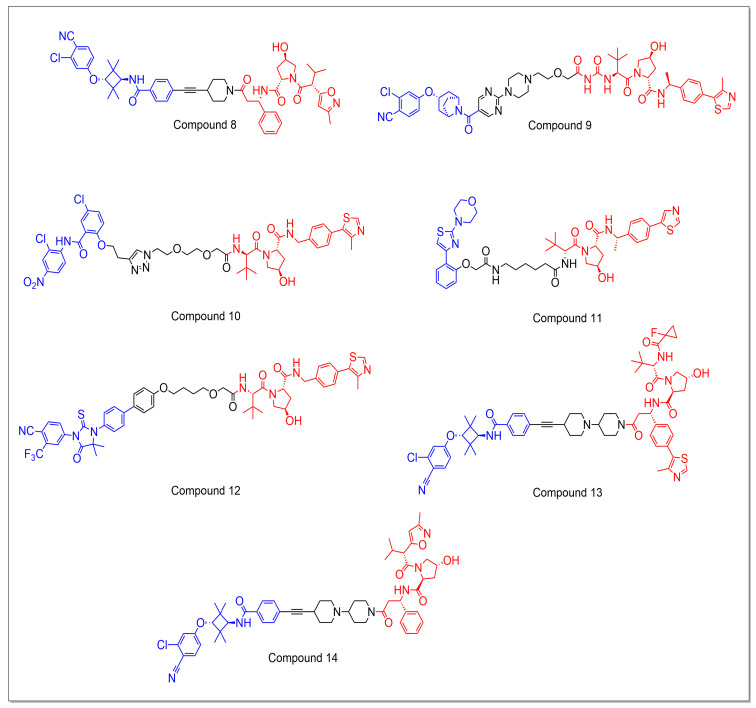
The chemical structures of VHL-based PROTACs that target the androgen receptor typically display the POI ligand in blue, the E3 ligase ligand in red, and the linker region in black.

**Figure 7 molecules-28-03698-f007:**
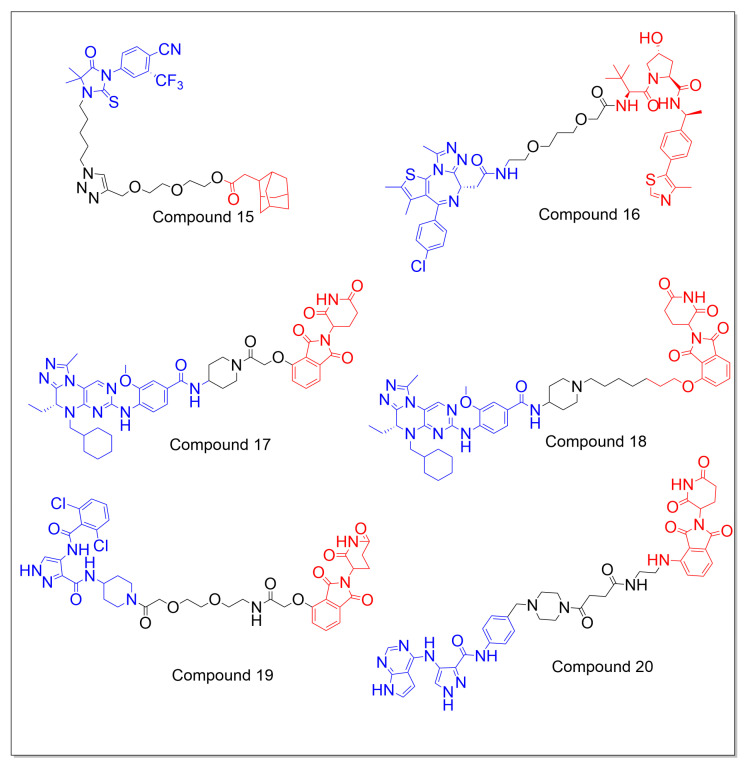
Chemical structures of miscellaneous E3 ligase ligand-based PROTAC (compound **15**)-targeting androgen receptor and PROTACs targeting non-AR targets (compound **16–20**). The POI ligand is shown in blue, the E3 ligase ligand is in red, and the linker is represented by black.

**Table 1 molecules-28-03698-t001:** PROTAC degraders in clinical trials for prostate cancer [3].

Name of the PROTAC	Clinical Trial No.	Date of Entry into Clinical Trials	Highest Clinical Phase	Target Protein	E3 Ligase Used
ARV-110	NCT03888612	March, 2019	Phase II	AR	Cereblon
CC-94676	NCT04428788	June, 2020	Phase I	AR	_
ARV-766	NCT05067140	October, 2021	Phase I	AR	Cereblon

**Table 2 molecules-28-03698-t002:** Notable differences between SMIs and PROTACs.

Parameter	SMIs	PROTACs
Druggability	Only limited targets are druggable	Extended scope for druggable targets
Selectivity	Lack of selectivity	Highly selective
Dosage	Require high dose; thus they are less potent	Minimal dose and highly protein
MOA	Competitive active site inhibition	Proteasomal degradation of target protein
Drug–target interactions	Usually they are non-covalent and require numerous interactions for good activity Require active site binding	POI requires minimal interactions and E3 ligase requires strong interactions for good activity No need for active site binding
Lipinski RO5	Most of them obeys RO5	Most of the PROTACs do not obey Lipinski rules
Synthetic feasibility	Usually easy to synthesize	Usually strenuous to synthesize

MOA—mechanism of action; POI—protein of interest; Ro5—rule of 5.

## Data Availability

Not applicable.

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
