# Peer review of "PROTACs in the Management of Prostate Cancer"

_molecules, 2023, doi:10.3390/molecules28093698_

Round 1

Reviewer 1 Report

This is an interesting paper reviewing the advanced literature on prostate cancer treatments by PROTACs. This kind of review was still missing in the international literature, therefore, I believe that it will be really appreciated from worldwide researchers.

Minor Points:

However, I have some suggestion to improve the quality of the paper.

Title: I would suggest converting “Protacs” in “PROTACs because it is acronymous.

Row 18: “Prostate cancer” – I would change “cancer” in “Cancer” such as is followed by the acronymous.

Rows 42-43: “permanently” – I can understand the sense of the sentence, but I would ever suggest to better explain that such as TPD systems are usually reversible.

Row 45: Related to “Small-Molecule PROTACs for Cancer Immunotherapy” (reference 4), the authors should cite this reference too:

-        Prozzillo, Y.; Fattorini, G.; Santopietro, M.V.; Suglia, L.; Ruggiero, A.; Ferreri, D.; Messina, G. Targeted Protein Degradation Tools: Overview and Future Perspectives. Biology 2020, 9, 421. https://doi.org/10.3390/biology9120421

Rows 179-180: It is not clear the meaning of these two rows (Top of form – Bottom of Form)

Row 372: “…androgen receptor…” Please, consider the full-length name only the first time you mention it in the text, use the acronymous thereby.

Rows 376, 492, 658-659: Please, indicate what stand for blu, black or red part of the molecules described in the figures.

Row 687: “…as a means of inducing protein degradation” – A reference would be appreciated at the end of the sentence.

Rows 693-694: “…targeted the AR. Non-AR 693 proteins also play…” – It is not clear to me. The full stop could be an error.

Rows 719-720: “...novel strategies like molecular glue, LYTAC, AUTAC, ATTEC, and RIBOTAC, which…” – I would suggest converting the sentence in “...novel strategies like molecular glue, LYTAC, AUTAC, ATTEC, and RIBOTAC, which…”. I would also recommend to insert references at the end of the sentence for these strategies.

Reviewer 2 Report

This review introduced PROTACs and highlighted the usage of PROTACs in managing prostate cancer. Overall, the whole manuscript is well structured, it can be accepted if the following comments are well addressed:

1.       In “2.1 Types of targeted protein degraders”, there are a variety of PROTACs mentioned here, it is nice to have citations for each PROTAC.

2.       From line 139 to line 142, These two sentences are repeated.

3.       In line 146, it’s clear to state “broken down by 26S proteasome” rather than “the 19s cap and 20s core of the 26s proteasome”.

4.       In line 152, it should be E2 ubiquitin conjugating enzyme, not binding enzyme.

5.       In line 163, I am lost in the first sentence, didn’t get what the authors want to express.

6.       In line 179, 180 and 357, “Top of form” and “Bottom of form” need to be deleted.

7.       In section 4, it is good to draw a figure to clearly present non-AR and AR related pathways.

Round 2

Reviewer 2 Report

my comments have been well addressed, it can be accepted in present form.